# Structural Optimization Design of a Six-Degrees-of-Freedom Serial Robot with Integrated Topology and Dimensional Parameters

**DOI:** 10.3390/s23167183

**Published:** 2023-08-15

**Authors:** Jiguang Jia, Xuan Sun

**Affiliations:** 1College of Mechanic and Control Engineering, Guilin University of Technology, Guilin 541004, China; jjg@glut.edu.cn; 2Key Laboratory of Advanced Manufacturing and Automation Technology, Education Department of Guangxi Zhuang Autonomous Region, Guilin University of Technology, Guilin 541006, China

**Keywords:** critical variables, integrated optimization method, stiffness–mass metamodel, MOGA

## Abstract

In the structural design of serial robots, topology and dimensional parameters design are independent, making it challenging to achieve synchronous optimization design between the two. To address this issue, a topology-and-dimension-parameter integrated optimization method (TPOM) is proposed by setting critical variables to connect topology layout and dimensional features. Firstly, the topology layout is extracted by the edge detection technique. Structural manufacturability reconstruction is conducted by measuring the dimensions of the layout through a program. Additionally, for the reconstructed structural layout, critical variables are set using three-dimensional software (SOLIDWORKS2021). The experiments primarily involve critical variables, quality, and deformation as variables. Then, the response surface methodology is selected to construct the stiffness–mass metamodel, and based on this, the structural deformation is analyzed. Lastly, the multi-objective genetic algorithm (MOGA) is employed to optimize the critical variables, and an optimized structure is established for validation. The results indicate that the proposed method (TPOM) reduces the mass of the structure by 15% while maintaining its stiffness. In addition, the deformation of the whole structure is less than 0.352 mm, which meets the requirements of industrial applications. Through quantitative analysis of the experimental results, the feasibility and superiority of the proposed method have been demonstrated.

## 1. Introduction

Robotics is a field that focuses on the development and study of robots, encompassing the modeling, design, and manufacturing of robotic systems [1]. Industrial robots are extensively employed in the automotive and electronics industries, as well as in other fields [2,3,4]. Serial robots [5,6,7] are essential components of robots. The complex parameter coupling and multi-objective design of serial robots have attracted the attention and research of scholars [8,9,10,11]. To address the structural requirements for serial robots in different environments, researchers evaluate and improve the performance of robots through simulations, providing accurate design and optimization solutions [12,13].

With the industry’s continuous improvement and the demand for precision component machining, serial robots face higher performance requirements, especially in structural design. In the past, the design approach involved increasing the cross-sectional area of the stress concentration regions in the structure to enhance stiffness. However, this design method increases the serial robots’ weight and causes more significant deformation under the influence of gravity. To address this issue, in recent years, new technologies such as topology optimization, evolutionary structural optimization (ESO), and solid isotropic material with penalization (SIMP) have been widely used in the structural design of serial robots. Liang, M. et al. [14] conducted topology optimization on the robots based on a rotor-torsion spring model and the finite element method. Yao, P. et al. [15] employed the Tosca module in ABAQUS to complete the structural topology optimization and improve the inherent frequency of the robots for the low-order case. Srinivas, G.L. et al. [16] used software to optimize the structure of the robot, aiming to achieve an optimal distribution of mass. Wang et al. [17] established a topology optimization system for the main components of the robot and constructed a stiffness–mass metamodel to obtain the robot’s optimization. Hegde et al. [18] optimized the finite element model to maximize the stiffness and minimize the structure’s mass. Guan et al. [19] utilized ADAMS and ANSYS software to optimize the structure of an industrial handling robot, enhancing its operational efficiency and performance. Yun Fei, B. et al. [20] accomplished the topology optimization of the upper arm structure of the robot using the SIMP method. Through expert analysis, topology optimization has been regarded as an ideal approach for lightweight structural design. Compared to simple modifications of existing models, topology optimization offers entirely new layout solutions, which the above expert research findings have confirmed. Although the above methods can optimize the robot, there has been a lack of in-depth research on optimizing dimensional parameters for the robot. To address the issue of parameter optimization, scholars have proposed various methods. Rout, B.K. et al. [21] utilized evolutionary optimization methods for structural optimization while concurrently designing the optimal parameters and tolerances for the robot arm. Hu, M. et al. [22] conducted multi-objective optimization of Cobots by selecting structural dimensions and parameterizing joint components as optimization variables. Liu et al. [23] designed the joint structure and linkages of the robot, identifying the parameters that influence its performance. Gupta et al. [24] researched the structure of serial robots and optimized it by treating the mass of the linkages as a design variable. Yin et al. [25] performed parameter optimization by considering factors such as structural dimensions, motors, and gearboxes as design variables. Zeinon, G. [26] and his colleagues optimized the use of DH parameters by analyzing the robot’s dynamics. Zhou et al. [27] proposed an integrated optimization method for designing lightweight robots. To optimize four parallel robots, Ben Hamida, I. et al. [28] employed a genetic algorithm to enhance their global flexibility and compactness. Chong, Z. et al. [29] developed a parameter model for worst-case identification of a 2-DoF hybrid robot. Crucial components were topologically optimized to reduce structural deformation. Although these methods address the optimization design issues of serial and hybrid robots, the serial robot’s performance is not limited to dimensional parameters. It is also significantly influenced by structural topology. To put it differently, the optimization of serial robots should consider both dimensional parameters and topological structures to achieve better performance. Kouritem, S. et al. [30] utilized stress analysis to determine the physical dimensions of the robot, thereby improving its performance. Hagenah, H. et al. [31] employed lightweight materials to manufacture advanced lightweight robots. Sahu et al. [32] utilized the finite element method (FEM) to determine the robot’s maximum allowable stress and deformation distribution. Liao et al. [33] analyzed the FRP lightweight robot’s geometric shape and manufacturing parameters, achieving optimized robot operation. To further enhance the robot’s performance, researchers have proposed a design approach utilizing the stiffness–mass metamodel and conducted structural topology optimization [34]. Structural topology optimization has become a commonly used design approach for serial robot optimization [10]. The research method above designs the structure aiming to reduce mass while also studying the deformation of the optimized model. However, the shape and dimension obtained from topology optimization are reconstructed within a rough range based on the computational results. Additionally, the structural design has yet to investigate the correlation between topology and dimensional parameters.

To address the integrated optimization problem between topology and dimensional parameters in serial robots, this study proposes TPOM. First, the lower arm component of the serial robots with the most adverse working conditions was selected as the research subject. The constraints and load conditions of this component were determined. Through topology iteration, the layout of the structure was obtained. Then, the dimension information from the topology layout was extracted for reconstructing the structure. Subsequently, the Box–Benhnken experimental design (BBD) [35] obtained components with different specifications. The stiffness–mass metamodel was established by utilizing ANSYS software to extract the stiffness matrix of the experimental components. Based on this, the deformation of the entire component was analyzed. Pearson correlation analysis used SPSS software to explore the relationship between critical variables and stiffness. Finally, the multi-objective genetic algorithm (MOGA) was used to optimize the critical variables. The structure was reconstructed based on the optimization results, and the structure was embedded in a stiffness–mass metamodel for analysis. The optimization structure was then validated, and the results indicate that the TPOM method effectively addresses the integrated optimization of serial robot components.

## 2. Structural Description and Research Methodology of Serial Robots

### 2.1. Description of the Structure

Nowadays, the coordinated handling of dual robots in factories has been gaining increasing attention. The handling workstation of the dual robots is shown in Figure 1. This workstation encompasses two serial robots which are capable of synchronized object manipulation. Each robot is constructed with a base, three connecting components, a wrist section, and an end effector, enabling it to execute various actions including handling, gripping, and rotation. The robot’s overall structure can be presented in terms of its sequence of structural links and joints. The joints are capable of exerting forces in specific directions to move the connected components to specific positions.

### 2.2. Research Methods

In this paper, when exploring the integrated optimization method (TPOM), the first assumption is that the combination of topology and parameter optimization can optimize the structure, and experiments are conducted to address this assumption. During the experimental phase, it is assumed that there is a positive correlation between the stiffness of the structure and the amount of deformation. Furthermore, it is assumed that structural optimization through key variables can achieve the desired outcome. This study aims to optimize the structure by integrating topology and dimensional parameters. First, topology optimization is performed. Based on program measurements, one should obtain the lengths of each edge in the topological layout. The critical variables are set, and the edge lengths of the topological layout are represented as a function of the critical variables. Subsequently, one should use the MOGA to optimize the critical variables to obtain the optimal parameters for the lower arm structure and validate the final structure. The research methodology is illustrated in Figure 2.

## 3. Structural Analyses

Among the components of serial robots, the lower arm is the main moving component, characterized by its significant weight and volume [15]. Due to the significant impact of the weight of the lower arm on the overall performance of the serial robot system, the TPOM method is employed to optimize the lower arm and enhance the overall performance. The lower arm can be considered a cantilever beam when the serial robot is in its most unfavorable working condition. In this study, a structural coupling method is utilized to couple all bolt surfaces and reference points, followed by the application of loads and boundary conditions. The schematic diagram of the analysis is shown in Figure 3.

The force and moment equilibrium analysis was conducted to determine the load distribution on the lower arm. Additionally, the stiffness matrix K of the lower arm components in the absolute coordinate system can be obtained through finite element software and structural stiffness analysis. Here, K is a 6 × 6 diagonal matrix where the values on both sides are equal. The main diagonal elements represent the stiffness values.
(1)$$K=\left[\begin{array}{cccccc} k_{11} & k_{12} & k_{13} & k_{14} & k_{15} & k_{16}\\ k_{21} & k_{22} & k_{23} & k_{24} & k_{25} & k_{26}\\ k_{31} & k_{32} & k_{33} & k_{34} & k_{35} & k_{36}\\ k_{41} & k_{42} & k_{43} & k_{44} & k_{45} & k_{46}\\ k_{51} & k_{52} & k_{53} & k_{54} & k_{55} & k_{56}\\ k_{61} & k_{62} & k_{63} & k_{64} & k_{65} & k_{66} \end{array}\right]$$

As serial robots are composed of links and joint structures, the overall flexibility of the robot can be represented as the flexibility of the links and joints. Here, the virtual joint method is considered for analyzing and describing the flexibility of the lower arm components, thereby obtaining the structural stiffness. The method involves incorporating an equivalent virtual spring at the end of the lower arm structure to represent the flexibility and coupling between different deflections of the lower arm components. The influence of gravity on the lower arm is neglected for the convenience of the study. In the analysis of the lower arm, it is assumed that each component is linearly elastic. According to Hooke’s law, in an elastic system, there is a linear relationship between the force on the virtual joint and the corresponding deflection. When the virtual joint is subjected to external forces, it generates reactive forces proportional to the deflection. This means that if the applied external force increases, the deflection of the virtual joint also increases accordingly.
(2)$$\left\{ \begin{array}{l} k_{pl}=diag\left(k_{1},\;k_{2},\;k_{3},\;k_{4},\;k_{5},\;k_{6}\right)\\ F_{pl}=k_{pl}\delta q_{pl}\;\\ F_{ol}=k_{ol}\delta q_{ol}\; \end{array}\right.$$
where $F_{pl}$ is a general joint force in the virtual joint, $k_{pl}$ is the diagonal matrix representing the coupling of force and deflection in the virtual joint, and $k_{1},\;k_{2},\;k_{3},\;k_{4},\;k_{5},\;k_{6}$ is the stiffness factor. $\delta q_{pl}\;$ is the corresponding joint deflection. $F_{ol}$ is the force on the links in the virtual joint, and $\delta q_{ol}\;$ is the corresponding link deflection. According to the constraint premise of neglecting gravity, it can be determined that:(3)$$F_{pl}=F_{ol},\delta q_{jpl}=\delta q_{pl}+\delta q_{ol}$$

Thus, the above equation can be re-expressed as:(4)$$\left\{ \begin{array}{l} \delta q_{jpl}=c_{jpl}F_{pl}=c_{jpl}F_{ol}\\ c_{jpl}=diag\left(c_{1},\;c_{2},\;c_{3},\;c_{4},\;c_{5},\;c_{6}\right) \end{array}\right.$$
where $\delta q_{jpl}$ is the virtual joint equivalent deflection, $c_{jpl}$ is the equivalent compliance matrix for the virtual joint l, and $\left(c_{1},\;c_{2},\;c_{3},\;c_{4},\;c_{5},\;c_{6}\right)$ is the equivalent flexibility factor of the virtual joint. The equivalent stiffness matrix for l is:(5)$$\left\{ \begin{array}{l} K_{jpl}=c_{jpl}^{-1}\\ c_{jpl}=c_{pl}+c_{ol}\\ \;\;=k_{pl}+k_{ol} \end{array}\right.$$
where $c_{pl}$ is the equivalent compliance matrix for the corresponding joint. Structural stiffness analysis and finite element analysis, both of which yield the stiffness matrix of the lower arm, allow for the evaluation of stiffness performance.

### 3.1. Topological Design of the Structure

In the optimization process, to obtain accurate computational results and reduce iterative computation time, pre-processing of the lower arm’s structure was conducted before experimentation. In practical operations, mesh refinement requires a balance between computational cost and accuracy. While meeting the requirements for solution accuracy and demands, the refined mesh size and type were determined. In this study, a mesh size of 8 mm was established, with a total of 51,159 mesh elements for the structure, and 101,336 nodes. Additionally, the Solid186 mesh type was employed, as it offers strong precision and stability, making it suitable for mesh partitioning of irregular geometric models.

The design domain for the lower arm topology was defined during the research process, as shown in Figure 4 below. By conducting an optimization design, a topological layout for the lower arm is obtained. In this process, the two end regions of the lower arm are designated as fixed mass regions and are not involved in the optimization design. In this process, the two end regions of the lower arm are designated as fixed mass regions and are not involved in the optimization design. Topology optimization of the components is performed using ANSYS software, employing the solid isotropic material with penalization (SIMP) method, with a boundary constraint of [0, 1]. The essence of the SIMP method lies in the penalization constraint imposed on material density. By introducing a penalization parameter into the optimization objective, the density is coupled with the material’s elastic modulus. As density approaches 0, the material’s elastic modulus increases, thereby reducing the load-bearing capacity of that region. When density approaches 1, the material’s elastic modulus retains its original value, representing a fully solid state. Topology optimization uses a solution method based on the optimality criterion algorithm. In this process, logical elements are used to evaluate the differences between different density parameters (x) before and after updating. The criterion for the iterative loop was such that the value of the difference in the convergence of the density parameters for all elements reached 0.1%.
(6)$$\left\{ \begin{array}{l} Y\left(x\right)=Y_{\min}+x^{p}\left(Y_{0}-Y_{\min}\right)\\ x\in \left[0,1\right] \end{array}\right.$$

In the above equation, Y(x) is Young’s modulus of elasticity and $Y_{0}$ represents the overall stiffness of the material. $x^{p}$ (p=3) denotes a penalty parameter for removing intermediate density values, and $Y_{\min}$ is the stiffness value of the void region.

In the design domain of structure, the density of each element is independently designed to alter the material distribution of the structure and generate a topological layout. To obtain the desired topological layout, iterative calculations are performed within the overall mass domain [0, 1] of the lower arm. Different mass retention rates (u) for the components are set, with a specified range of mass variation of u = 0.05. The objective function is set as the compliance (C) of the component, and the mathematical model for topological optimization is expressed by Equation (10).
(7)$$Find\;x={\left(x_{1},x_{2},\ldots \ldots x_{n}\right)}^{T}$$
(8)$$\min:C(x)=U^{T}KU$$
(9)$$S\mathrm{ubject}\;\mathrm{to}.\;\left\{ \begin{array}{l} V(x)-V\times u=0\\ F=KU\\ 0<x_{\min}\leq x_{i}\leq 1\left(x=1,2\ldots \ldots n\right) \end{array}\right.$$
where $x_{i}$ is the design variables, and $x_{\min}$ is the minimum relative density. V(x) is the final structural volume. U is the displacement vector, and u is the different mass retention rates. F is the external load, K is the global stiffness matrix, and V is the pre-optimization structural volume. A “program-controlled” solver suitable for shell and elongated body models is selected during the topological optimization process to obtain the structural layout. The topological result of the lower arm after iterative calculations is shown in Figure 5.

### 3.2. Obtaining Dimensional Parameters Using a Measuring Program

After iterative calculations, the topology structure exhibits irregular shapes and rough edges, making it unsuitable for direct manufacturing. A new method is proposed to make the post-topology dimensions as close as possible to the actual results. This method is based on edge detection and utilizes a measurement program to obtain the dimensions of the topological layout. Edge detection primarily involves the following steps: (1) binary processing of the topological image; (2) calculation of multiple gradients using the Sobel operator, followed by weighted summation; (3) normalization of the convolution results to obtain an edge map; (4) identification of regions in the image where pixel values undergo significant changes, enabling the detection and extraction of these edge features; and (5) computation of gradient magnitudes for each pixel.

By following these steps, edge detection can be performed on the topological layout image, allowing the extraction of edge features. The calculation of gradient magnitude for each pixel in the image is as follows:(10)$$\left\{ \begin{array}{l} Gx=\left(P\left(x+1,\;y-1\right)+2P\left(x+1,\;y\right)+P\left(x+1,\;y+1\right)\right)\;\\ -\left(P\left(x-1,\;y-1\right)+2P\left(x-1,\;y\right)+P\left(x-1,\;y+1\right)\right)\\ Gy=\left(P\left(x-1,\;y+1\right)+2P\left(x,\;y+1\right)+P\left(x+1,\;y+1\right)\right)\\ -\left(P\left(x-1,\;y-1\right)+2P\left(x,\;y-1\right)+P\left(x+1,\;y-1\right)\right) \end{array}\right.$$
(11)$$G\left(x,y\right)=\sqrt{{\left(Gx\right)}^{2}+{\left(Gy\right)}^{2}}$$

In the above equation, Gx denotes the gradient value of the pixel in the horizontal direction, and Gy  denotes the gradient value of the pixel in the vertical direction. P(x,y) denotes the pixel value at the image position (x,y). G(x,y) is the gradient magnitude. Afterward, a measurement program was developed to extract dimensionality from the layout. The core idea of this program was to measure the coordinates of two pixels in the image using a callback function. These two points were considered intersection points, and the dimension between them was calculated using the ‘math.dist’ function. Then, this line was displayed on the image. By utilizing these dimension parameters, errors in the reconstruction process can be minimized. The process is illustrated in Figure 6.

### 3.3. Manufacturable Reconfiguration of Structure

Utilizing the measurements obtained from the lower arm’s topological layout, the structure has been reconstructed to ensure manufacturability in accordance with processing requirements. The symmetrical nature of the lower arm is evident on both sides. As a result, it is only necessary to reconstruct the top and bottom features of the component, along with one side feature. The manufacturability reconstruction is shown in Figure 7, where the red lines indicate the dimensions between two tangent points. The pattern formed by these red lines represents the layout that needs to be reconstructed. The influence of fillets is not considered in this process. Before the manufacturability reconstruction of the model, the structure of the lower arm was irregular and contained many burrs, which posed significant difficulty for processing. After the manufacturability reconstruction, the lower arm structure exhibits complete features.

## 4. Stiffness–Mass Metamodel

### 4.1. Setting of Critical Variables and Experimental Design

In the study mentioned above, the layout of the topological structure was obtained. To further investigate the relationship between dimensional parameters, mass, and structural stiffness, critical variables were defined for the dimension parameters of the topological layout. This process removed small layout features and rounded corners from the model. The design set $S=\left\{s_{1},s_{2},s_{3},\ldots ,s_{n}\right\}$ was defined, in which each component in set $s_{1},s_{2},s_{3},\ldots ,s_{n}$ represented dimension parameters. Based on the topological layout, the critical variable $N=\left(n_{1},n_{2},n_{3},n_{4},n_{5},n_{6}\right)$ was set for the lower arm. The setting process is illustrated in Figure 8. Subsequently, the critical variables were treated as design parameters. The outcome was achieved without loss of generality, where the critical variables ranged:$$\left\{ \begin{array}{l} n_{1}=\left(36.9,45.1\right),\;n_{2}=\left(46.8,57.2\right),\;n_{3}=\left(31.5,38.5\right)\\ n_{4}=\left(64,76\right),\;n_{5}=\left(121.5,148.5\right),\;n_{6}=\left(100.8,123.2\right) \end{array}\right.$$

The Box–Benhnken (BBD) [36] experimental design was selected for the study. BBD design offers advantages such as a reduced number of experiments, good design efficiency, and robustness. To investigate the relationship between critical variables, mass, and structural stiffness, a total of 49 experimental groups were designed. The experimental groups are presented in Table 1, below.

### 4.2. Stiffness–Mass Metamodel

This study establishes critical variables based on the theory of dimensional parameters and topological structure. To deeply investigate the effect of dimensional parameters on stiffness, the critical variables, mass, and stiffness are introduced into the RSM [37] and FEA software to establish the mapping of the experimental model. Equation (16) represents the relationship between critical variables, the stiffness in three directions, and the mass of each component group.
(12)$$\left\{ \begin{array}{l} diag(K_{I})=f(N)\\ m_{I}=f(N) \end{array}\right.$$
where $K_{I}$ is the stiffness matrix of the component, and I and N are the crucial variables. A polynomial (e.g., Equation (17)) is used to construct the RSM, which describes the mapping relationship between stiffness and mass. Here, a quadratic polynomial is utilized in creating the RSM.
(13)$$f(N)=\alpha_{0}+\sum_{i=1}^{t}b_{i}n_{i}+\sum_{i=1}^{t}c_{i}n_{i}{}^{2}+\sum_{i=1}^{t}\sum_{i<r}^{t}d_{ir}n_{i}n_{r}+\sum_{i=1}^{t}s_{i}n_{i}{}^{3}$$

In the above equation, $n_{i}$ and $n_{r}$ are the critical variables in N. $n_{i}{}^{2}$, $n_{i}{}^{3}$ denote second- and third-order nonlinearities. $n_{i}n_{r}$ is the interaction term for any two parameters, and $\alpha_{0}$, $b_{i}$, $c_{i}$, $d_{ir}$, $s_{i}$ is the regression coefficient for each term. The BBD experimental design was established in the study above, in which each experimental group variable corresponded to a mass. The stiffness matrix corresponding to the mass was obtained by analyzing and calculating each experimental group. The stiffness–mass metamodel of the component is shown in Figure 9, below.

In Figure 9, $Lk_{1}$ and $Ak_{1}$ represent the linear stiffness and angular stiffness of each experimental group, respectively. Among them, the structures of experimental groups 17 and 44 as to design domains is shown in Figure 10. According to the results, it was found that the X-, Y-, and Z-directional stiffness values of the lower arm component generally exhibit consistent trends. As the mass increases, the stiffness may decrease, indicating a nonlinear relationship between mass variation and overall stiffness. For the non-cuboid structure of the lower arm, the linear stiffness is highest on the Z-axis and lowest on the X-axis, in contrast to the angular stiffness. The angular stiffness of the lower arm reaches its maximum along the X-axis and its minimum along the Z-axis. The study reveals that the stiffness along the Y-axis falls between the values observed along the X and Z axes. Moreover, the stiffness values in these three directions do not influence one another, and no noticeable pattern or regularity emerges among the stiffness values in different directions, indicating significant differences.

In the context of vertical comparisons, the study reveals that experimental groups with varying dimensional parameters display distinct trends in stiffness and mass alterations. There appears to be a correlation between maximum mass and minimum stiffness. Similarly, linear stiffness variations along the X, Y, and Z directions exhibit consistent trends in transverse comparisons. However, angular stiffness demonstrates dissimilar variations. An increase in angular stiffness along the X-axis corresponds to a decrease in angular stiffness along the Y-axis and Z-axis. Furthermore, the stiffness displays significant discrepancies and abrupt shifts in structures sharing similar dimensional parameters. Building upon the stiffness–mass metamodel, the deformations of each group’s critical variable structure were investigated. Figure 11 illustrates the deformations in the X, Y, and Z directions.

According to the results shown in Figure 11, among all components, the lower arm has the largest deformation on the X-axis and the smallest deformation on the Z-axis. The deformation on the Y-axis follows a similar trend as the stiffness changes, lying between the X and Z axis. Furthermore, the results confirm the strong deformation resistance of the parts within the design ranges of critical variables. The finite element analysis was performed in Workbench, where the deformation of the part was calculated by setting the material type and using SOLID186 for meshing.

### 4.3. Correlation Analysis

In this paper, Pearson correlation analysis [38] is used to explore the relationship between independent variables and dependent variables. Here, the critical variables were used as independent variables, and the stiffness values in the X, Y, and Z directions were studied as dependent variables to explore the relationship between each group of critical variables and stiffness. Pearson correlation coefficients (r) were calculated. When the *p*-value between two variables is less than 0.05, it indicates statistically significant evidence of a relationship between them. The strength of the correlation was then determined based on the magnitude of the correlation coefficient (r). When −1≤r≤+1, the larger the absolute value of r, the stronger the correlation between the variables. The analysis revealed a significant correlation between variables $n_{4}$ and $n_{6}$ and the stiffness values in all three directions, while the remaining variables showed a weak correlation. The analysis results are shown in Figure 12.

Based on the above analysis, it can be concluded that the strongest correlation with stiffness is observed between variables $n_{4}$ and $n_{6}$, while the remaining variables show weaker correlations with stiffness. The variable is the most important factor affecting the stiffness in the X-direction, while the remaining variables have similar degrees of impact on stiffness in all three directions. The variable has a smaller impact on stiffness compared to variables $n_{4}$ and $n_{6}$. Importantly, the stiffness values in the X, Y, and Z directions largely adhere to this pattern.

## 5. Optimized Designs

### 5.1. Design for Component Optimization

The optimization goal for the lower arm component is to achieve high stiffness (i.e., minimal deformation) while reducing weight. Therefore, a response surface was established to model the relationship between critical variables, mass, and deformation. A quadratic polynomial approximation model, as expressed in Equation (13), was used to construct the response surface. Based on this approximation model, multi-objective optimization using MOGA was performed. To ensure the adequacy of the approximation model, performance metrics were employed to assess the fitting accuracy of the quadratic polynomial fit. The metrics encompassed in this set were the average relative mean absolute error (Average), maximum relative absolute error (Maximum), root mean square error (RMSE), and $R^{2}$, as outlined in Equations (14) and (15).
(14)$$\left\{ \begin{array}{l} \mathrm{Average}=\frac{\sum_{b=1}^{h}|y_{b}-{\hat{y}}_{b}|}{\sum_{b=1}^{h}|y_{g}-{\bar{y}}_{b}|}\\ \mathrm{Maximum}=\frac{\max\{|y_{b}-{\hat{y}}_{b}|,\ldots ,|y_{b}-{\hat{y}}_{b}|\}}{\sum_{b=1}^{h}|y_{b}-{\bar{y}}_{g}|/h} \end{array}\right.$$

In the above equation, $y_{b}$ is the actual value mapped for the experimental group I, ${\hat{y}}_{b}$ is the predicted value for the experimental group, and ${\bar{y}}_{b}$ is the mean value. h is the set of evaluation parameters.
(15)$$\left\{ \begin{array}{l} \mathrm{RMSE}=\sqrt{\frac{\sum_{b=1}^{h}{\left(y_{b}-{\bar{y}}_{b}\right)}^{2}}{h}}\\ R^{2}=1-\frac{\sum_{b=1}^{h}{\left(y_{b}-{\hat{y}}_{b}\right)}^{2}}{\sum_{b=1}^{h}{\left(y_{b}-{\bar{y}}_{b}\right)}^{2}} \end{array}\right.$$

A value closer to 1 indicates higher fitting accuracy. Conversely, a minor root mean square error (RMSE) is desirable. The results (Figure 13) demonstrate excellent fitting accuracy. Specifically, $R^{2}$ values are greater than 0.9, RMSE is less than 0.2, the average value is less than 0.2, and the maximum value is less than 0.3. M represents the fitting accuracy for mass, while D represents the fitting accuracy for deformation. Due to the overall high fitting accuracy, the fitted quadratic polynomial can be used as the objective function expression for optimization design.

During the optimization process, the critical variables are treated as design variables, while the objective functions are the mass and deformation of the lower arm component. The mathematical expression and constraints for the objective function are as follows:(16)$$\min M_{p},\mathrm{and}\;\min D$$
(17)$$\left\{ \begin{array}{l} s.t\;m_{u}{+m}_{l}\leq m_{t}\;\\ D\leq 0.352\mathrm{mm}\\ T\left(x\right)\geq 0\\ n^{l}\leq n\leq n^{u} \end{array}\right.$$
where $M_{p}$ is the mass of the lower arm part, and T(x) is the geometric constraints, including the mass of the part, and the values of the critical variables. $n^{l}$, and $n^{u}$ are the upper and lower limits of the variables. The multi-objective genetic algorithm (MOGA) [39,40,41] optimization aims to obtain a set of solutions, known as the Pareto frontier. After the MOGA optimization of the lower arm component, a set of non-dominated Pareto front solutions is obtained. To find the best compromise solution, a cooperative equilibrium is employed to obtain the final results. This process is similar to a cooperative game in which participants strive to maximize their interests while considering the information and interests of other participants. In this cooperative game, the cooperative equilibrium point is the solution closest to the ideal point and represents the best compromise solution among the participants.
(18)$$f_{i}\left(x\right)=\frac{f_{ij}\left(x\right)-k_{f}{}_{i}}{\sigma_{fi}}$$
(19)$$D_{c}=\sqrt{\sum_{\tau =1}^{3}{\left(f_{c,\tau }\left(x\right)-f_{\tau ,\min}\right)}^{2}}$$

In the above equation, $f_{ij}\left(x\right)$ and $\sigma_{fi}$ are the mean and variance of the optimization objective, and $f_{c,\tau }\left(x\right)$ is the value of the objective for the cth Pareto point. $f_{i}\left(x\right)$ is the ith objective and $f_{\tau ,\min}$ is denoted as the minimum value of the τth objective. The minimum point $D_{c}$ is the solution on the Pareto frontier closest to the ideal solution. In other words, among all possible solutions, it aligns with the ideal point and exhibits the best overall performance.

### 5.2. Results of Optimization

In this study, the MOGA (multi-objective genetic algorithm) and cooperative equilibrium principles were employed to select critical variables for the lower arm component. The MOGA method combines the characteristics of genetic algorithms and multi-objective optimization techniques. To ensure algorithm convergence, an initial sample size of 6000 was established, generating 1200 samples at each iteration. The best individual accounted for 0.7 of the population, and the maximum number of iterations was set at 20. A subset of the optimization results constituted the Pareto frontier, which was subsequently analyzed using a dimensionless approach for objectives on this frontier. This approach enabled the determination of the optimal distance to the ideal point. Furthermore, the concept of cooperative equilibrium was utilized to identify a state that closely approached the ideal value through a balanced consideration of various interests. The cooperative equilibrium points for mass and deformation were 15.854 kg and 0.282 mm, respectively. The optimization results for the critical variables are presented in Figure 14 and Table 2, respectively.

### 5.3. Validation of Optimization Results

This paper conducted a parameterized search for all combinations of critical variables of the lower arm component. Post-optimization, the mass of the lower arm was determined to be 15.854 kg, with a deformation of 0.282 mm. Furthermore, to exhibit the efficacy of the TPOM method, the lower arm component underwent a reconstruction process for validation. All preprocessing steps, including variable sizing, design objectives, and constraints, remained consistent throughout the design process. Under identical constraints, the validation outcomes demonstrate the viability of utilizing the TPOM method for optimizing the component. The stiffness characteristics and deformations across the three directions for the optimized part using TPOM are detailed in Table 3. The results demonstrate that the optimized structure obtained through TPOM reduced the mass from 18.65 kg to 15.854 kg while exhibiting superior stiffness performance. Additionally, the total deformation was only 0.282 mm.

Specifically, under the same constraints, the stiffness values in the three directions of the lower arm component significantly impact the overall deformation. The total deformation of the lower arm varies with changes in the stiffness values in the X, Y, and Z directions. Notably, the stiffness in the Z-direction is more significant than in the other directions, and therefore the deformation in that direction is smaller. The deformation of the initial structure (IS) of the lower arm, the topologically optimized structure (TS), and the TPOM structure (TPOMS) are depicted in Figure 15.

The total deformation of the IS, TS, and TPOIMS was 0.224 mm, 0.577 mm, and 0.282 mm, respectively. By the usage requirements specified by the factory, the total deformation of the lower arm must not exceed 0.352 mm. Notably, the deformation achieved through TPOM is considerably smaller than the prescribed limit, underscoring the effectiveness of the proposed method. The study reveals that under load conditions, deformations in the X, Y, and Z directions calculated using the IS and TPOMS models are significantly less pronounced than those in the TS structure. Among these directions, the Y direction exhibits the highest degree of deformation. While the TS design records a total deformation of 0.577 mm, surpassing the stipulated usage threshold, IS and TPOMS exhibit total deformations of 0.224 mm and 0.282 mm, respectively. It is imperative to emphasize that excessive deformation may lead to structural impairment. Consequently, practical applications of serial robotic lower arms experience bending deformations. The overall deformation of the lower arm structure designed by topology reaches 0.577 mm, indicating a lack of corresponding bending strength in this design. Compared to IS, after an optimized design using TPOM, the mass of the lower arm is reduced by 15%, and the maximum overall deformation is only 0.282 mm, which meets the usage requirements. The stiffness of the lower arm component is shown in Figure 16.

To further investigate the TPOM, Figure 14 compares the structures’ linear stiffness and angular stiffness. The results show the following: (1) The linear stiffness values in the X, Y, and Z directions tend to be equal, with little variation in fluctuations. (2) Among the angular stiffness values in the X, Y, and Z directions, the angular stiffness along the Z-axis is the highest. The TS exhibits the lowest angular stiffness of all three directions, contributing to the excessive deformation. (3) TPOM optimization reduces the mass of the parts by 15%, and the linear stiffness values in the X, Y, and Z directions are approximately equal to that of the IS.

## 6. Conclusions

In the past, the optimization of serial robot structures primarily relied on topology optimization to reduce weight, without considering the influence of dimension parameters on the structure. Additionally, a majority of studies have overlooked the stiffness of the optimized structure and failed to delve into the relationship between design variables, mass, and stiffness. The integrated topology-and-dimensional-parameter optimization method (TPOM) proposed in this paper aims to accomplish the optimal design of the structure using the mechanism theory and engineering experience of both. This study employs critical variables to integrate the topology structure and dimension parameters for optimal design. The proposed new method (TPOM) successfully resolves the synchronization issue between topology structure and dimension in the design process. This method has been validated through the optimization of the lower arm component. The conclusions are as follows:(1)A topological design domain for components was established, and a measurement program based on edge detection was employed to obtain the topological layout, thereby avoiding manual errors associated with direct model reconstruction.(2)A stiffness–mass element model was established, and the overall deformation of the member was analyzed. Subsequently, the correlation between critical variables and stiffness was studied, revealing an influential relationship between them.(3)The TPOM approach integrates topology optimization and dimension optimization. This was validated using an example of the lower arm component. The results demonstrated that TPOM reduces the component’s mass while satisfying stiffness requirements, thereby proving its effectiveness and advantages.

The lower arm component designed by the proposed method will be fabricated in the future. Then, the effect of the overall performance of the serial robot will be tested, and the effects of the topological shape, dimensional parameters, structure, and other factors of the robot components will be further investigated.

## Figures and Tables

**Figure 1 sensors-23-07183-f001:**
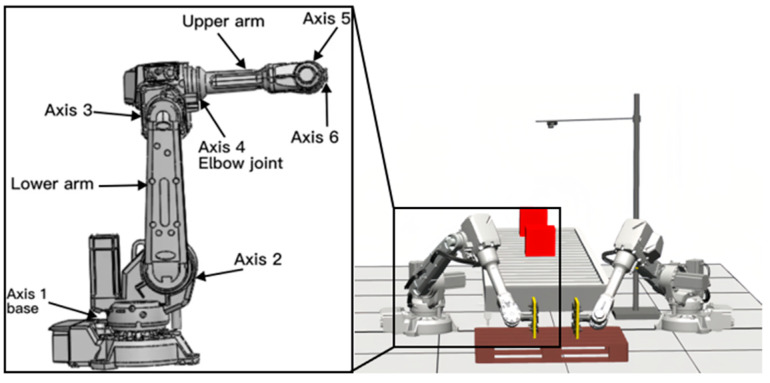
Structure of serial robots.

**Figure 2 sensors-23-07183-f002:**
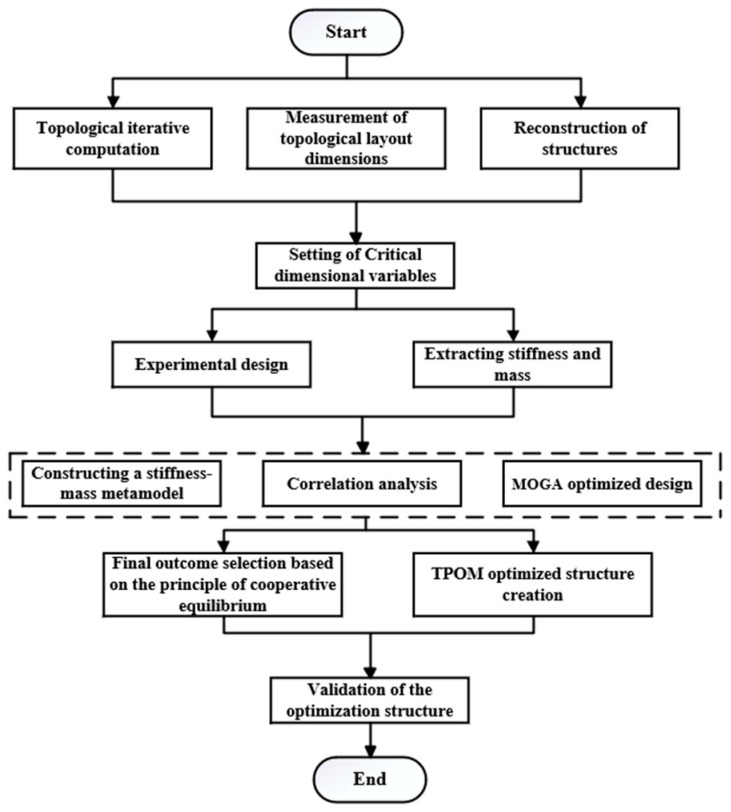
The research methodology.

**Figure 3 sensors-23-07183-f003:**
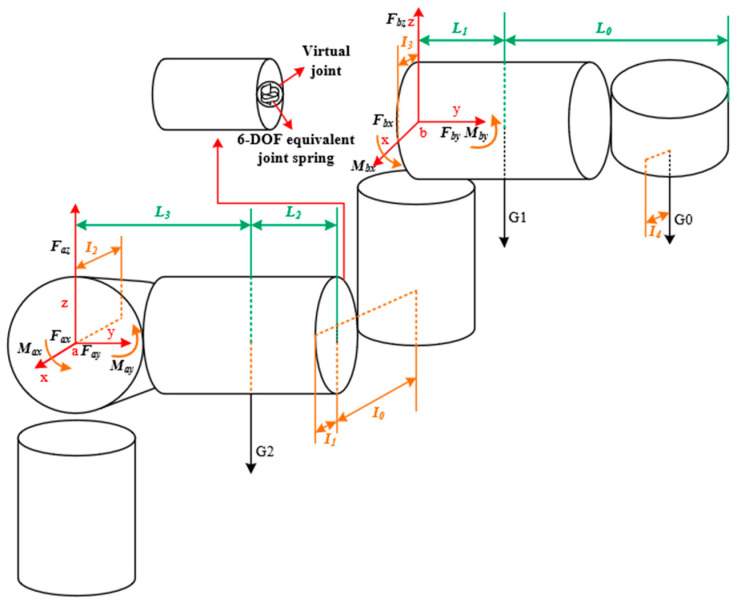
Constraints and loads applied to the lower arm.

**Figure 4 sensors-23-07183-f004:**
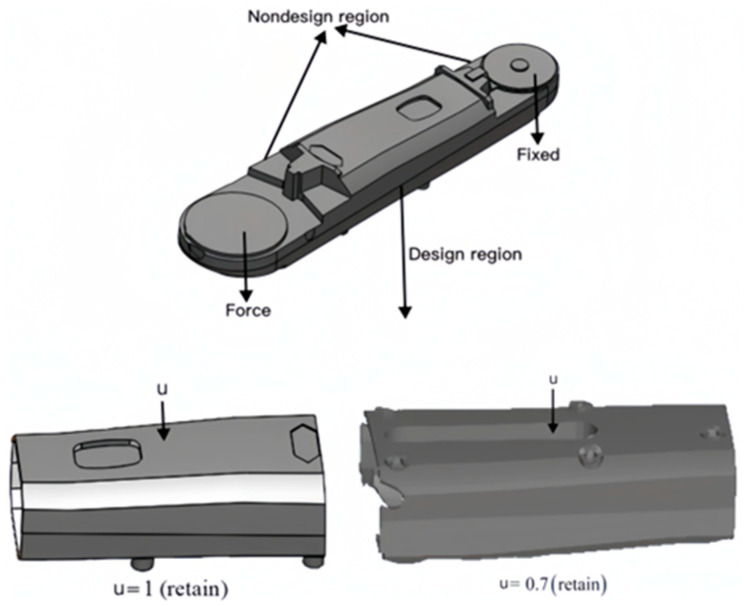
Topology design domain.

**Figure 5 sensors-23-07183-f005:**
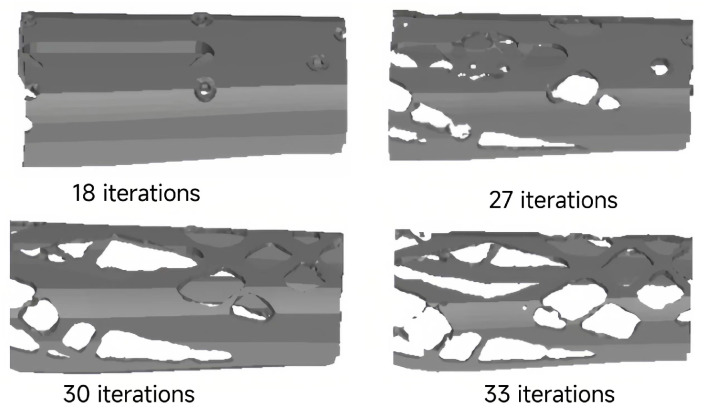
The layout of the structure after the topology optimization.

**Figure 6 sensors-23-07183-f006:**
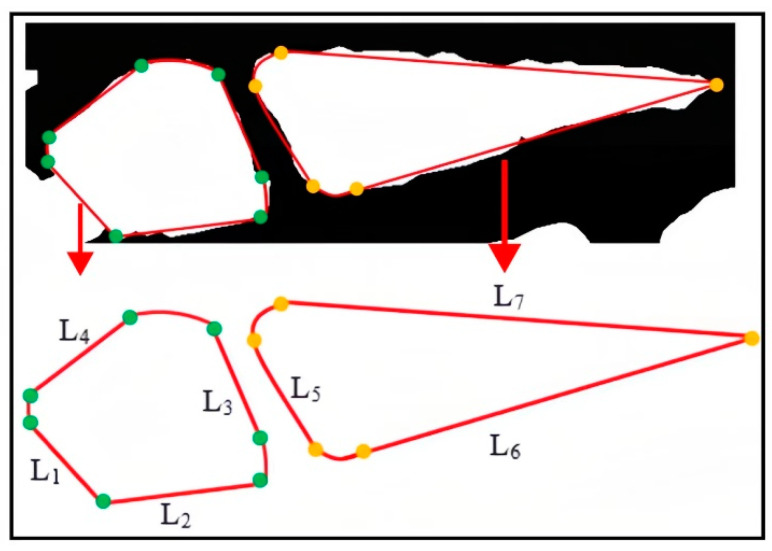
Schematic diagram of the extraction topology layout dimension. In this diagram, $L=\left(L_{1},L_{2},L_{3},L_{4},L_{5},L_{6},L_{7}\right)$. L is the pixel distance between the two points, which is the dimension distance between the two endpoints.

**Figure 7 sensors-23-07183-f007:**
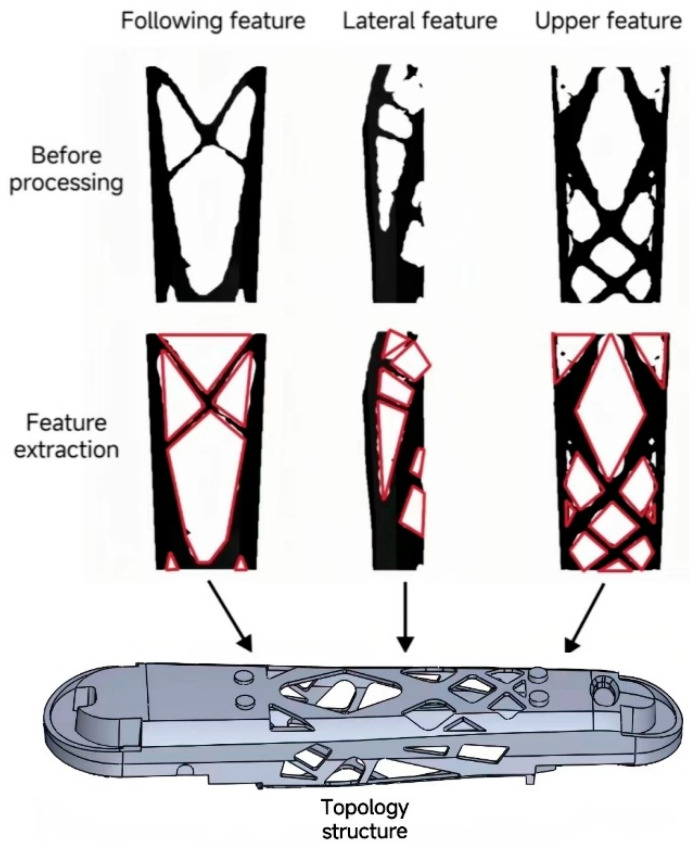
Manufacturable reconstruction for topology optimization.

**Figure 8 sensors-23-07183-f008:**
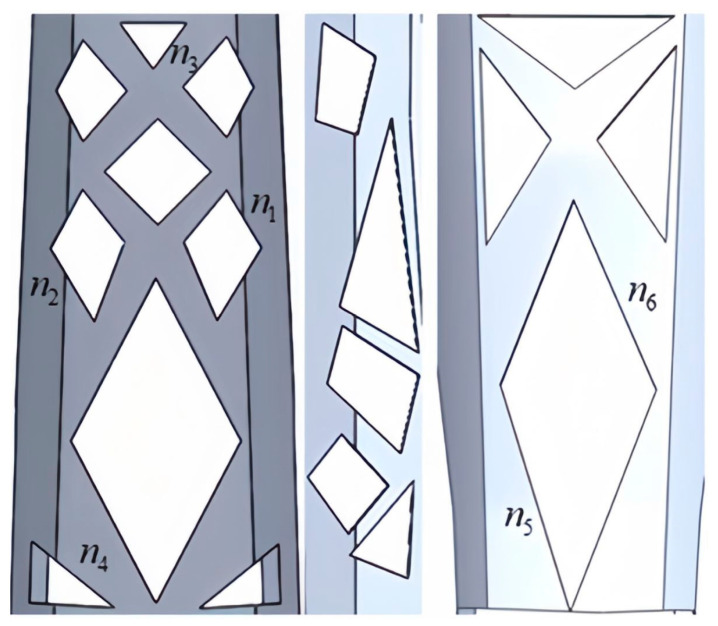
The setting of critical variables.

**Figure 9 sensors-23-07183-f009:**
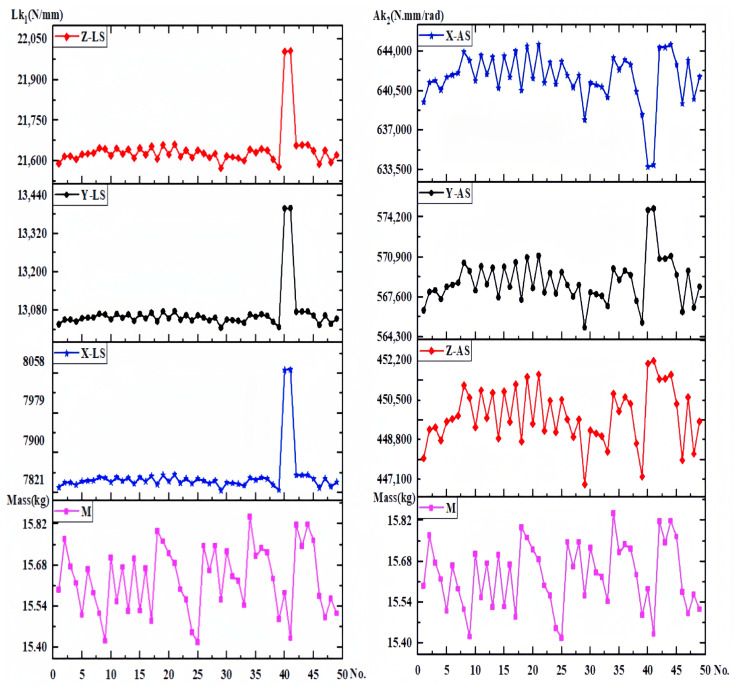
Stiffness–mass metamodel.

**Figure 10 sensors-23-07183-f010:**
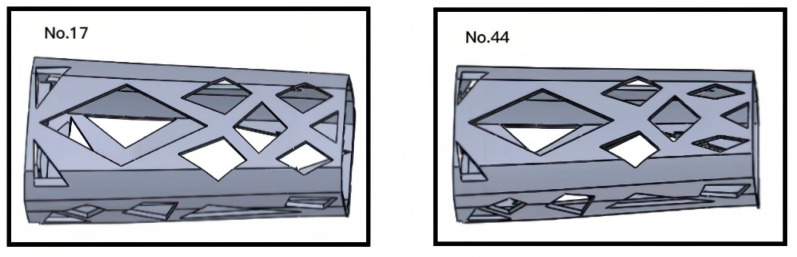
The design domain structure of the experimental group.

**Figure 11 sensors-23-07183-f011:**
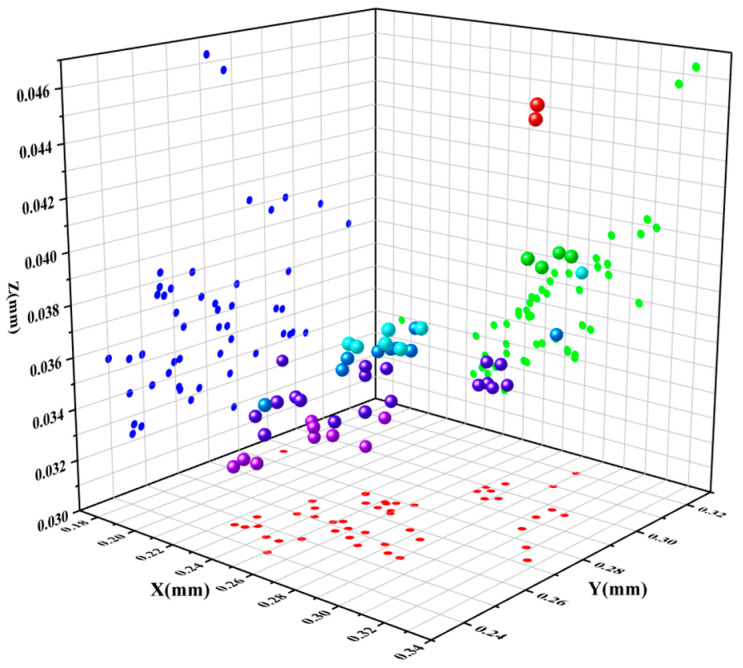
Deformation in three directions.

**Figure 12 sensors-23-07183-f012:**
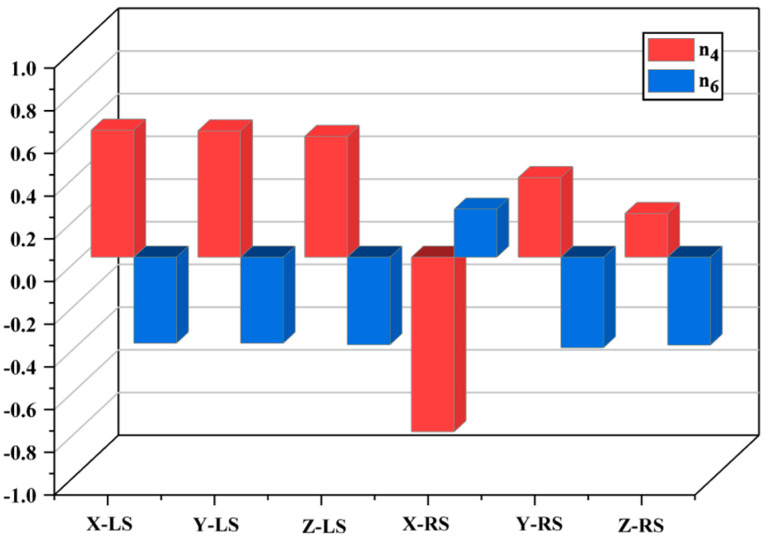
Analysis results.

**Figure 13 sensors-23-07183-f013:**
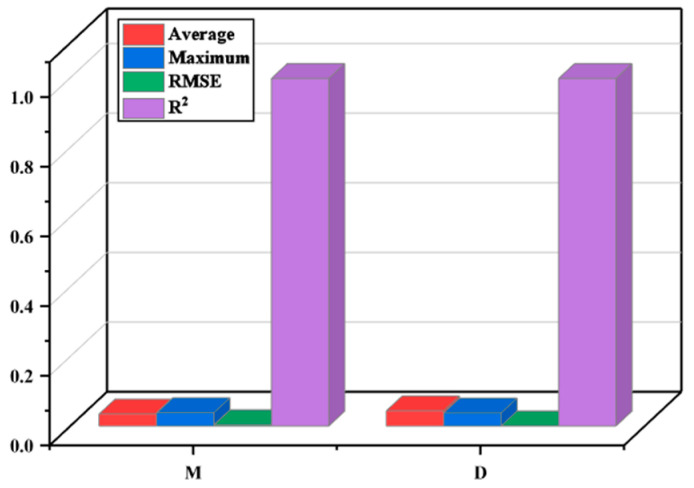
Fitting accuracy of the approximate model.

**Figure 14 sensors-23-07183-f014:**
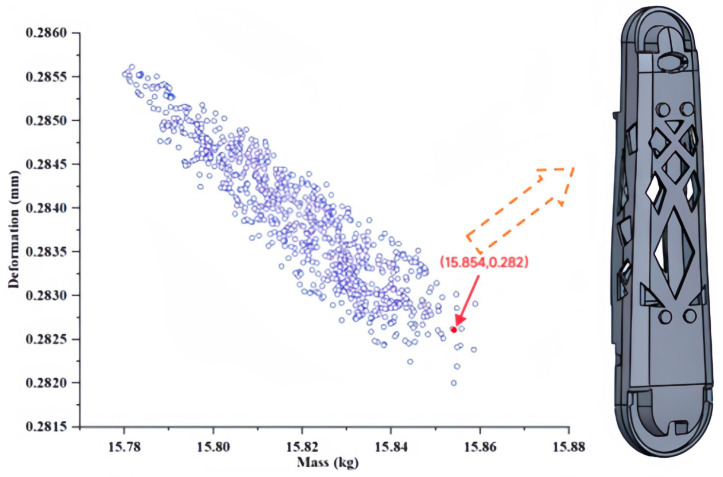
Optimization results.

**Figure 15 sensors-23-07183-f015:**
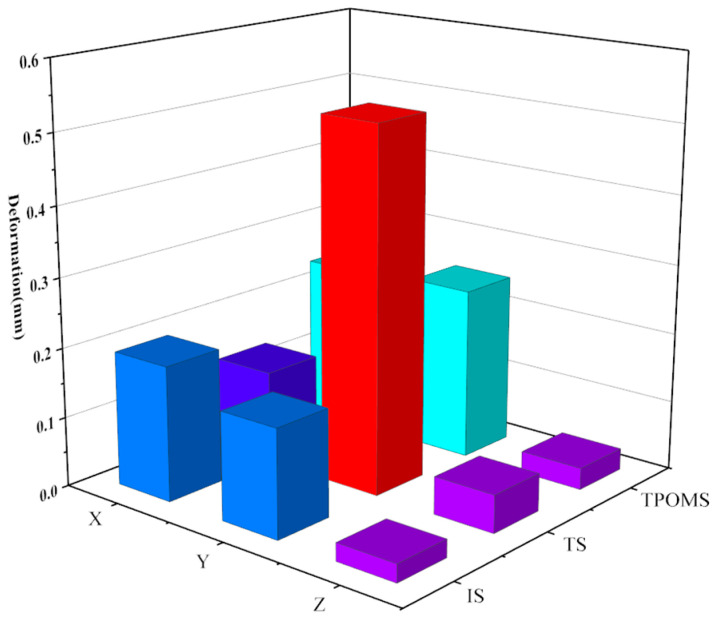
Deformation of the IS, TS, and TPOMS.

**Figure 16 sensors-23-07183-f016:**
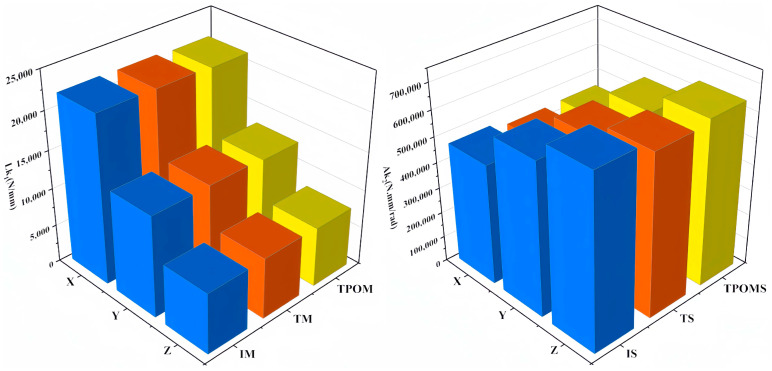
Comparison of stiffness results for different structures of lower arm parts.

**Table 1 sensors-23-07183-t001:** Experimental design of BBD.

Name	$n_{1}$	$n_{2}$	$n_{3}$	$n_{4}$	$n_{5}$	$n_{6}$	$n_{1}$	$n_{2}$	$n_{3}$	$n_{4}$	$n_{5}$	$n_{6}$
1	41	52	35	70	135	112	36.9	52	35	64	121.5	112
2	36.9	46.8	35	64	135	112	45.1	52	35	64	121.5	112
3	45.1	46.8	35	64	135	112	36.9	52	35	76	121.5	112
4	36.9	57.2	35	64	135	112	45.1	52	35	76	121.5	112
5	45.1	57.2	35	64	135	112	36.9	52	35	64	148.5	112
6	36.9	46.8	35	76	135	112	45.1	52	35	64	148.5	112
7	45.1	46.8	35	76	135	112	36.9	52	35	76	148.5	112
8	36.9	57.2	35	76	135	112	45.1	52	35	76	148.5	112
9	45.1	57.2	35	76	135	112	41	46.8	35	70	121.5	100.8
10	41	46.8	31.5	70	121.5	112	41	57.2	35	70	121.5	100.8
11	41	57.2	31.5	70	121.5	112	41	46.8	35	70	148.5	100.8
12	41	46.8	38.5	70	121.5	112	41	57.2	35	70	148.5	100.8
13	41	57.2	38.5	70	121.5	112	41	46.8	35	70	121.5	123.2
14	41	46.8	31.5	70	148.5	112	41	57.2	35	70	121.5	123.2
15	41	57.2	31.5	70	148.5	112	41	46.8	35	70	148.5	123.2
16	41	46.8	38.5	70	148.5	112	41	57.2	35	70	148.5	123.2
17	41	57.2	38.5	70	148.5	112	36.9	52	31.5	70	135	100.8
18	41	52	31.5	64	135	100.8	45.1	52	31.5	70	135	100.8
19	41	52	38.5	64	135	100.8	36.9	52	38.5	70	135	100.8
20	41	52	31.5	76	135	100.8	45.1	52	38.5	70	135	100.8
21	41	52	38.5	76	135	100.8	36.9	52	31.5	70	135	123.2
22	41	52	31.5	64	135	123.2	45.1	52	31.5	70	135	123.2
23	41	52	38.5	64	135	123.2	36.9	52	38.5	70	135	123.2
24	41	52	31.5	76	135	123.2	45.1	52	38.5	70	135	123.2
25	41	52	38.5	76	135	123.2						

**Table 2 sensors-23-07183-t002:** MOGA optimization results.

Critical Variables (cm)						Mass (kg)	Total Deformation (mm)
$n_{1}$	$n_{2}$	$n_{3}$	$n_{4}$	$n_{5}$	$n_{6}$	M	D
45.072	49.921	34.527	70.663	134.77	102.58	15.854	0.282

**Table 3 sensors-23-07183-t003:** Performance of the model after TPOM optimization.

Category	Linear Stiffness (N/mm)	Angular Stiffness (N.mm/rad)	Deformation (mm)
X	7864.61	646,069.03	0.24
Y	13,090.53	572,303.96	0.25
Z	21,683.59	451,792.87	0.03

## Data Availability

Important data are contained within the article. Additional data may be available upon reasonable request to the corresponding author.

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
