# Peer review of "Structural Optimization Design of a Six-Degrees-of-Freedom Serial Robot with Integrated Topology and Dimensional Parameters"

_sensors, 2023, doi:10.3390/s23167183_

Round 1

Reviewer 1 Report

In this article, the author designed and applied topology optimization for dimensional parameters . A topology and dimension parameter integrated optimization method (TPOM) is used for this purpose.

Please list the assumptions followed in the framework.

Please specify the nomenclature in a  separate section

The authors are suggested to review some literature on topology/structural optimization and existing methods like SIMP in the CAD/CAE packages.

Briot, Sébastien, and Alexandre Goldsztejn. "Topology optimization of industrial robots: Application to a five-bar mechanism." Mechanism and Machine Theory 120 (2018): 30-56.

Dara, Ashok et al., "Does topology optimization exist in nature?." National Academy Science Letters 45.1 (2022): 69-73.

Zhu, Benliang, et al. "Design of compliant mechanisms using continuum topology optimization: A review." Mechanism and Machine Theory 143 (2020): 103622.

The methodology has to be presented clearly.

please explain How the mesh refinement is done for the optimal solution

please give appropriate figure numbers with detailed labels  (for Figure 16)

Do a comparative assessment with recent literature to draw the merits of the proposed work over the existing methods.

Author Response

Dear Reviewer:

Thank you for your comments concerning our manuscript entitled “Structural Optimization Design of a Six Degrees of Freedom Serial Robot with Integrated Topology and Dimensional Parameters”. Your comments are all valuable and helpful for revising and improving our paper, as well as providing important guidance for our research. We have carefully studied your suggestions and made revisions to the manuscript. Please see the attached document for our responses.

Sincerely,

Xuan SUN

Reviewer 2 Report

The paper is well structured. The objective is clearly described in the introductions, and the conclusions are supported by the results.

Even when the method is well described, there are some points that should be described.

First of all, the arm of the robot has been optimized. The topology optimization method usually requires the discretization of the domain in finite elements. There should be a brief paragraph concerning the FEM model: number of elements, element type (solid element, shell, etc.) or computation time.

The second point is the analysis of the results. In Figure 5 the evolution of a optimized layout is shown. This structure exhibits very thin bar, as consequence of the optimization process. Is well known that there are some methods to control the minimum length scale in these problems: robust formulation, control of the perimeter, use of PDF filtering, among others. It is a good idea to use any of this algorithms to avoid this thin part. Also, the authos indicates that a linear analysis is performed to obtain the displacements, but with kind of structures (again, with thin bars) a buckling anlysis is recommended in order to see the behaviour of the structure.

Another question is the postprocess implemented to ensure the manufacturability of the design, because probably, with a control of the minimum length scale of the solid region (black) the manufacturability is facilitated.

Minor revision is required.

Author Response

(The authors gave the same response as above.)

Round 2

Reviewer 1 Report

The manuscript is well revised, and no further comments to authors.

All the best!.